PlasmidSeeker: identification of known plasmids from bacterial whole genome sequencing reads

Roosaare Märt 1 mrt.roos@gmail.com
http://orcid.org/0000-0001-6268-037X Puustusmaa Mikk 1
Möls Märt 1 2
Vaher Mihkel 1
http://orcid.org/0000-0003-3966-8422 Remm Maido 1
1 Department of Bioinformatics, IMCB, University of Tartu , Tartu , Estonia
2 Institute of Mathematics and Statistics, University of Tartu , Tartu , Estonia
Gillespie Joseph
Electronic publication date: 2018 Apr 2
Publication date: 2018
Volume: 6
Electronic Location ID: e4588
Received 2018 Jan 26; Accepted 2018 Mar 18
Copyright: © 2018 Roosaare et al.
Copyright year: 2018
Copyright holder: Roosaare et al.
License: This is an open access article distributed under the terms of the Creative Commons Attribution License, which permits unrestricted use, distribution, reproduction and adaptation in any medium and for any purpose provided that it is properly attributed. For attribution, the original author(s), title, publication source (PeerJ) and either DOI or URL of the article must be cited.
License URL: https://creativecommons.org/licenses/by/4.0/

Keywords: Plasmid, k-mer, Whole genome sequencing, Unassembled

Funding: Estonian Centre of Excellence in Genomics and Translational Medicine 2014-2020.4.01.15-0012 Estonian Ministry of Education and Research IUT34-11 This work was supported by the European Union through the European Regional Development Fund through Estonian Centre of Excellence in Genomics and Translational Medicine (project No. 2014-2020.4.01.15-0012) and by the Estonian Ministry of Education and Research (institutional grant IUT34-11). There was no additional external funding received for this study. The funders had no role in study design, data collection and analysis, decision to publish, or preparation of the manuscript.

==============================
Background

Plasmids play an important role in the dissemination of antibiotic resistance, making their detection an important task. Using whole genome sequencing (WGS), it is possible to capture both bacterial and plasmid sequence data, but short read lengths make plasmid detection a complex problem.

Results

We developed a tool named PlasmidSeeker that enables the detection of plasmids from bacterial WGS data without read assembly. The PlasmidSeeker algorithm is based on k-mers and uses k-mer abundance to distinguish between plasmid and bacterial sequences. We tested the performance of PlasmidSeeker on a set of simulated and real bacterial WGS samples, resulting in 100% sensitivity and 99.98% specificity.

Conclusion

PlasmidSeeker enables quick detection of known plasmids and complements existing tools that assemble plasmids de novo. The PlasmidSeeker source code is stored on GitHub: https://github.com/bioinfo-ut/PlasmidSeeker.

Introduction

Plasmids are circular or linear double-stranded DNA molecules capable of autonomous replication and conjugation. They have been described in all three domains of life (Antipov et al., 2016). Bacterial plasmids often confer beneficial traits to their hosts, such as antimicrobial resistance, which has directly contributed to the rapid increase of multidrug-resistant bacteria, described as one of the greatest dangers to human health (Nyberg et al., 2016). Thus, considerable effort has been put into plasmid detection and monitoring.

There are several methods for plasmid detection, all with their own merits and drawbacks. PCR-based replicon typing targets the conserved replicon sites of plasmids (Smalla, Top & Jechalke, 2015) and it can be expanded to target many replicons by using multiplex PCR (Carattoli et al., 2014). While quick and cheap, multiplex PCR is hard to extend to cover all novel plasmid groups (Carattoli et al., 2014). Pulsed field gel electrophoresis reveals the size and number of plasmids in an isolate, but the process can take several days (Nyberg et al., 2016). Moreover, neither of these methods yields much information about the plasmid DNA sequence. In recent years, optical DNA mapping of plasmids has been developed, based on the visualization of the plasmid DNA stretched in nanofluidic channels. With the help of fluorescent dyes, a unique barcode roughly depicting the DNA sequence of a plasmid can be made and compared to barcodes in a reference database. While very promising, optical mapping may not yet be suitable for the detection of shorter (<50 kbp) plasmids (Nyberg et al., 2016).

Whole genome sequencing (WGS) has been extensively used for studying bacterial isolates in both clinical and research setting as it can now be obtained within short timescales and at relatively low cost (Orlek et al., 2017). The majority of bacterial WGS projects consist of short Illumina reads, making plasmid detection a complex task that requires specialized bioinformatics tools. Carattoli et al. (2014) developed plasmidfinder web tool that searches for conserved replicon sites using blastn and compares them to a curated database of plasmid replicons. PlasmidSPAdes (Antipov et al., 2016) uses the read coverage of contigs to distinguish between plasmid and bacterial sequences. PLACNET (Lanza et al., 2014) on the other hand utilizes info from both coverage and reference plasmids and outputs a plasmid network. All these tools require the assembling of raw reads into contigs.

At present, the National Center of Biotechnology Information (NCBI) RefSeq database contains over 8,000 complete plasmid sequences, due to the rapid development of next-generation sequencing technologies in the last decade. With such a large and constantly growing database, it is plausible to perform quick monitoring for known (reference) plasmids rather than full-scale de novo assembly of plasmids in the sample.

In this work, we describe PlasmidSeeker, a tool to detect known plasmid sequences from unassembled raw WGS reads, also eliminating the need for the read alignment process. PlasmidSeeker is based on the assumption that the copy number of a plasmid, therefore also coverage, is different (usually higher) compared to the chromosomal sequence of the bacterial isolate (Providenti et al., 2006). This resembles the approach of plasmidSPAdes, but instead of analyzing read coverage of contigs, which requires assembling, PlasmidSeeker is based on short DNA oligomers with length k (k-mers) and uses k-mer abundance to distinguish between plasmid and chromosomal k-mers. Although PlasmidSeeker is only able to detect reference plasmids without further sequence information, it is suitable as a first step in the analysis of plasmid content and complements other bioinformatics tools meant for assembling plasmids de novo.

Implementation

PlasmidSeeker consists of a reference plasmid database and a search algorithm. The database, consists of k-mer list files for each reference plasmid and a text-format index file connecting the name of each plasmid to its k-mer list. FASTA identifiers are used as plasmid names and can be changed in the index file as needed. The required input for building a database is a multi-FASTA file with plasmid sequences, which is also the format that can be downloaded from the RefSeq database. All operations with k-mers are performed with the GenomeTester4 toolkit (Kaplinski, Lepamets & Remm, 2015). Input is a FASTQ file containing raw WGS reads and a FASTA file containing the assembled genome of a reference bacterial strain related to the isolate. The search algorithm is outlined below.

The input sample file (raw WGS reads) is converted to a k-mer list and all singleton k-mers (k-mers that occurred only once) are eliminated as these are mostly due to sequencing errors.

The algorithm finds the approximate genome coverage of the isolated bacterium. For this, an assembled genome sequence of a bacterium must be provided by the user in FASTA format. The reference sequence should be as close as possible to the sequenced isolate. The genome coverage of the isolate is roughly equal to the median abundance of k-mers shared between the isolate and the reference bacterium (chromosomal k-mers), assuming that most of the k-mers in the bacterial genome are unique (occur only once in the sequence).

A fraction of detected unique plasmid k-mers F is analyzed for all reference plasmids. Only reference plasmids with F above a threshold (default 80%) are analyzed further and reported in the output.

The average plasmid copy number per bacterial cell is estimated by dividing the median k-mer abundance of the given plasmid with the median k-mer abundance of chromosomal k-mers.

Similar plasmids, determined by the overlap coefficient C (fraction of shared k-mers relative to the smaller plasmid), are clustered together in the results as a single hit if C exceeds a threshold (default 80%). Output is a tab-delimited text file.

Materials and Methods

Plasmid database

All 9,351 available plasmid sequences were downloaded in April 2017 from the NCBI RefSeq database. As the dataset also contained partial plasmid sequences and antimicrobial resistance genes, we filtered it to only include sequences whose identifiers contained “plasmid” and not “gene,” “partial,” “incomplete” or “putative”. The downloaded multi-FASTA file was separated into the individual FASTA files of each plasmid, which were converted to k-mer lists using the GenomeTester4 toolkit. A text-format index file was created that connects the names of plasmid k-mer list files to their FASTA identifiers. The final plasmid database contained 8,514 plasmid sequences. The database built with k = 20, used in most of the tests, is available from https://figshare.com/s/5f7b924544839f7d6e59 or http://bioinfo.ut.ee/plasmidseeker/; uncompressed size on the disk is 8.8 GB.

Generating simulated WGS samples for optimal k-mer length selection

Four samples were generated, each containing reads from a bacterium (Escherichia coli O157:H7 str. Sakai (NC_002695.1), Pseudomonas aeruginosa UCBPP-PA14 (CP000438.1), Acinetobacter baumannii YU-R612 (CP014215.1), Staphylococcus aureus subsp. aureus MRSA252 (BX571856.1)) and 1–3 plasmids (Table 1). The sample of S. aureus was used as a negative control and was without plasmids. MetaSim, a next-generation sequencing simulator (Richter et al., 2011), was used to generate synthetic 80 bp reads (MetaSim cmd -mg errormodel-80bp.mconf -f [clone length] -r [number of reads]). Empirical error model for 80 bp reads was downloaded from https://ab.inf.uni-tuebingen.de/software/metasim/errormodel-80bp.mconf. In case of plasmid sequences shorter than 3,000 bp, clone length was set to 500. A total of 500,000 reads were generated for each plasmid and 1,000,000 for each bacterial genome. Each simulated sample consisted of 1,000,000 bacterial reads and a corresponding number of plasmid reads to ensure specific plasmid copy number in the sample. The number of plasmid reads added was calculated as follows: (number of bacterial reads × read length ÷ bacterial full genome length) × (desired plasmid copy number × plasmid length ÷ read length).

Table 1 Plasmid copy numbers predicted by PlasmidSeeker.

Strain	Plasmid	Plasmid length (bp)	Real copy number	Predicted copy number	
Pseudomonas aeruginosa (sim)	pNOR-2000	21,880	20.00	19.50	
pUM505	123,322	5.00	5.38	
pKLC102	103,532	5.00	5.75	
Acinetobacter baumannii (sim)	pABIR	29,823	10.00	10.15	
Escherichia coli (sim)	pOSAK1	3,306	10.00	10.33	
pO157	92,721	2.00	2.11	
Providencia stuartii	P. stuartii strain ATCC 33672 plasmid	48,844	3.02	2.89	
Citrobacter freundii	pKEC-a3c	272,297	5.23	5.17	
pCAV1335-5410	5,410	14.00	14.00	
Corynebacterium callunae	pCC1	4,084	11.75	11.90	
pCC2	82,716	0.87	0.90	
Note:

“Sim” represents a simulated sample. For real WGS samples, real copy number is given as estimated by Antipov et al. (2016). For real copy number calculations in simulated samples, see “Materials and Methods”.

Generating simulated WGS samples for mutational analysis

Eight samples were made with MetaSim (settings as above), each containing 80 bp reads from a bacterium (E. coli or P. aeruginosa, same as above) and a plasmid: pOSAK1 (3,306 bp) in case of E. coli and pUM505 (123,322 bp) in case of P. aeruginosa. Random substitution mutations were induced to plasmid reference sequences using an in-house script (PlasmidSeeker GitHub, “make_mutations.pl”). Average mutation rates per bp were as follows: 1/1,000, 1/300, 1/100, 1/30. Relative copy numbers of the plasmid and the bacterium were 10 to 1 in all samples, respectively. Number of bacterial reads in all samples was 1,000,000.

Generating simulated WGS samples for detectable k-mer fraction analysis

Sixteen samples were made with MetaSim (settings and error profile as above), each containing 80 bp reads from a bacterium (E. coli, P. aeruginosa, S. aureus or A. baumannii, same as above). Number of reads in each sample was equal to the respective genome length divided by the read length (80 bp) times the desired coverage (1, 3, 5 or 7). Theoretical distribution was assumed to follow Poisson distribution, read length was equal to the read length in simulated samples (80 bp) and the average error rate was 0.01/bp.

Escherichia coli samples and analysis with PlasmidSeeker and plasmidSPAdes

We used three E. coli WGS samples with sequence type 131 (Accession numbers as follows: EC1, ERR1937840; EC2, ERR1937914; EC3, ERR1937841; see also Table S3). PlasmidSeeker was run with F and C thresholds set to 80%. PlasmidSPAdes was run with default settings.

Statistical test for comparing isolate and plasmid copy numbers

A part or the whole plasmid can integrate into the bacterial genome. Additionally, bacterial genomes might contain k-mers that are also present in plasmids, just by chance. Therefore, the bacterial isolate genome may contribute k-mers to the fraction of plasmid k-mers. If the copy number of a plasmid is not significantly different from the chromosomal copy number, most of the detected plasmid k-mers might originate from the bacterial genome. Therefore, to detect whether k-mers are actually from a plasmid or from the bacterial chromosome, we test the hypothesis: H0: Covbacteria=Covplasmid

H1: Covbacteria≠Covplasmid

Covbacteria is the expected coverage of unique chromosomal k-mers and Covplasmid is the expected coverage of unique plasmid k-mers. We assume that the copy number of the plasmid is usually different from the copy number of the bacterial chromosome. Therefore, the expected coverage of plasmid k-mers is also different from the expected coverage of chromosomal k-mers. Hence, accepting the hypothesis H1 leads to the conclusion that at least some of the reads containing reference plasmid k-mers come from a plasmid.

To test the hypothesis, we fitted a mixture of negative binomial distributions to describe the distribution of the k-mer frequencies (we used the mixture distribution to allow some k-mers to be missing or to have increased copy number as k-mers might originate from repeat regions). We used separate mixtures to describe the plasmid and chromosomal k-mer frequencies. The method used assumes at least 70% of k-mers to be unique in the sequenced isolate and plasmid. We fitted two models by the maximum likelihood method. In one model, we restricted the expected coverage of unique k-mers to be the same for both plasmid and chromosomal k-mers. The second model allows the expected coverages to be different. We used a likelihood ratio type test to compare the two models. Copy number was considered significantly different at threshold p < 0.05 corrected with Bonferroni in case of multiple tests.

Results

Selecting an optimal k-mer length

First, we analyzed the effect of k-mer length on the uniqueness of chromosomal k-mers (k-mers that occur only once in the full chromosomal sequence) and on the fraction of k-mers shared between plasmids and chromosomes. The isolate genome coverage estimation can be biased by many multi-copy k-mers, thus it is essential to use k-mer lengths that are mostly unique in the bacterial genome. Plasmid k-mers that are shared with chromosomal k-mers can hinder plasmid detection. To analyze how k-mer length affects the fraction of unique chromosomal k-mers and the fraction of unique k-mers shared between a bacterial isolate and the plasmid database, we used full genome sequences of four clinically relevant bacterial species (Ventola, 2015): E. coli, A. baumannii, P. aeruginosa and S. aureus. Plasmid database consisted of 8,514 plasmid sequences downloaded from NCBI RefSeq database (see Materials and Methods). K values in range between 8 and 32 were tested. As expected, k-mer length has a strong effect on the fraction of unique chromosomal k-mers, until it reaches a plateau (Fig. 1, dashed lines). For all strains, the fraction of unique k-mers reaches plateau at k = 16, but the shared k-mer fraction reaches plateau at k = 20 (Fig. 1, solid lines). Thus, the optimal k-mer length should be at least 20.

Figure 1 Fraction of unique k-mers in bacteria and unique chromosomal k-mers shared with the plasmid database.

Dashed lines indicate the fraction of unique k-mers in the assembled full genome of the bacterium (number of unique k-mers divided by all k-mers of the bacterium), and solid lines indicate the fraction of unique chromosomal k-mers that are also present in the plasmid database.

Bacterial WGS data is unlikely to contain plasmids with 100% identical sequences to reference plasmids due to the sequence variation of plasmids and their ability to switch out entire gene islands (Couturier et al., 1988). A point mutation in a sequence would eliminate up to k unique k-mers. N point mutations can remove up to n × k k-mers if the distance between mutations exceeds k. Longer deletions d bp long, such as a lost gene island, remove approximately d k-mers. A shorter k-mer should therefore be more sensitive in detecting plasmids with mutations.

We assessed how mutations in a plasmid sequence affect the fraction of plasmid k-mers detected (F) by using simulated samples created with MetaSim (Richter et al., 2011) (see Materials and Methods): E. coli with the small, 3 kbp plasmid pOSAK1 (Fig. 2, solid lines) and P. aeruginosa with the large, 120 kbp plasmid pUM505 (Fig. 2, dashed lines). The database used in this and the following tests consisted of 8,514 plasmids. Expectedly, in all cases, increased rate of point mutations decreased F. Also, F decreased with increasing k-mer length, differences being larger in case of higher mutation rates. Overall, point mutations and longer deletions in plasmids have a negative impact on the fraction of plasmid unique k-mers detected in the sample. Longer k-mers are less suitable for the detection of plasmids that are more distantly related to the reference sequences. Tests show that short k-mers are better able to detect plasmids remotely related to database reference sequences. Taking the previous analysis and this into account, we chose k = 20 as the default value.

Figure 2 Fraction of detected plasmid k-mers (F) is affected by the distance from the reference plasmid and k-mer length.

F decreases with increasing k-mer length, with more pronounced differences with larger distances. In all cases, increasing the distance decreased F. Name, size and bacterial host of the plasmid were as follows: pUM505, 123,322 bp, P. aeruginosa; pOSAK1, 3,306 bp, E. coli. Green lines show test results for k = 16, orange line k = 24 and blue line k = 32. Distance from the reference plasmid is given in nucleotide substitutions per bp.

K-mer frequencies are affected by the coverage of the sample. In samples with higher coverages, most of the unique k-mers are present in frequency >1. PlasmidSeeker discards singleton k-mers in the search process as many of them are due to sequencing errors. However, if sample coverage is low, a significant fraction of chromosomal k-mers might be present only in a single copy. In this case, results can be largely biased. To find the lower limit for the sample coverage, we created simulated samples of E. coli, P. aeruginosa, S. aureus and A. baumannii with nucleotide coverages ranging from 1 to 7 (see Materials and Methods) and analyzed them with the PlasmidSeeker. Each sample was converted to a k-mer list and only k-mers not present in the genome of the reference bacterium, either E. coli or P. aeruginosa, were discarded. With 1× sample coverage, less than 20% of k-mers were detectable in sample (frequency >1; Fig. 3). The fraction of detectable k-mers increased with increasing sample coverage. Taking these observations into account, we recommend analyzing isolates with at least 5× coverage. This coverage would allow detection of single-copy plasmids.

Figure 3 Sample coverage affects the fraction of detectable (frequency >1) chromosomal k-mers in simulated samples.

Each simulated sample was converted to a k-mer list, and all k-mers not present in the reference bacterium were discarded. The fraction of detectable k-mers was calculated by dividing the number of k-mers with frequency >1 by the total number of strain k-mers. The theoretical distribution was assumed to follow a Poisson distribution, read length was equal to the read length in simulated samples (80 bp) and the average error rate was 0.01/bp.

Selecting a threshold value for the fraction of unique plasmid k-mers detected (F) from the sample

As multiple plasmids in the database can contain the same k-mer just by chance, PlasmidSeeker algorithm uses a threshold for the minimum required fraction of plasmid k-mers detected from the sample (F) to filter out as many false positives as possible. F cannot be too strict, otherwise small plasmids with only a few mutations would be filtered out as well (Fig. 2), resulting in false negatives.

In order to determine how different F threshold values affect the results, we used two datasets (Table 1, also see Materials and Methods). The first contained four simulated samples with 1–3 plasmids: E. coli, A. baumannii, P. aeruginosa and a negative control (S. aureus without plasmids). The second dataset, also used by Antipov et al. (2016) in benchmarking plasmidSPAdes, contained real WGS reads of Providencia stuartii, Citrobacter freundii, Burkholderia cenocepacia and Corynebacterium callunae, all with completed reference genomes and annotated plasmids. B. cenocepacia was used as a negative control in this set as it did not contain any plasmids. PlasmidSeeker was tested with five F thresholds ranging from 35% to 95% (Fig. 4). According to the previous test results, k-mer length was set to 20. As the number of true negatives (with 8,514 plasmids in the database) was very large, specificity was 99.98% or better in all cases. Also, with no false negative results in any tests, sensitivity was 100%. There were no false positives in case of P. stuartii, C. callunae and B. cenocepacia (Table S1). In other samples, false positives occurred when using F threshold values lower than 80%. The large amount of false positives in case of C. freundii are due to a lot of reference plasmids being similar to the two C. freundii plasmids. Taking into account that the tested samples contained only annotated plasmids, we recommend to keep F threshold value between 80% and 90% as 95% might be too strict and miss plasmids that are only remotely related to references (Fig. 2). In the clustering and E. coli tests below, we used 80% to achieve higher sensitivity.

Figure 4 Threshold of the fraction of plasmid k-mers detected (F) affects the number of false positives in real and simulated samples.

Number of false positive identifications decreased with higher F thresholds. At an F threshold of 95%, there were no false positives. No false negatives were detected at any threshold value, and there were no false positives for P. stuartii and C. callunae. Simulated samples are marked “sim”. Read length in simulated samples is 80 bp; C. callunae, 300 bp; C. freundii, 400 bp; P. stuartii, 202 bp.

Prediction of plasmid copy number

PlasmidSeeker estimates plasmid copy number per bacterial cell by dividing the median k-mer abundance of the given plasmid with the median k-mer abundance of chromosomal k-mers. Predictions were validated using either known copy numbers in case of simulated samples or copy numbers estimated by plasmidSPAdes in case of the previously mentioned benchmarking dataset. PlasmidSeeker copy number predictions were accurate in all cases (Table 1).

Clustering

PlasmidSeeker does not assemble reads, avoiding the possibility for errors in this step. Still, it cannot distinguish between very similar plasmids, any of which could be present in the sample as they all pass F threshold. Such plasmids are clustered together in the output, based on the overlap coefficient C (fraction of shared k-mers relative to the smaller reference plasmid) between k-mer lists (Fig. 5). Clustering algorithm is similar to the CD-HIT algorithm (Li & Godzik, 2006): plasmids are first sorted according to their lengths and the longest plasmid is chosen as the representative for the first cluster, recruiting other plasmids if C is over a threshold. Then, the process starts again from the longest plasmid not yet assigned to a cluster and continues until all plasmids are clustered (Fig. 5). Each reported cluster indicates that a plasmid from this cluster is present in the sample. Plasmids whose copy numbers do not have statistically significant difference from the chromosomal copy number are not included in the clustering step, but reported as single hits with their respective p-values.

Figure 5 Clustering algorithm used by PlasmidSeeker.

First, plasmids are sorted by length, starting with the longest (1). Clusters are formed based on the overlap coefficient C (fraction of shared k-mers relative to the smaller reference plasmid). In step 1, the longest reference plasmid from the results (its k-mer list) is picked and compared to all other detected reference plasmids. All plasmids with C exceeding a threshold are recruited to Cluster 1. In step 2, reference plasmids already placed to Cluster 1 are excluded. The process continues until all plasmids are assessed. Numbers depict different plasmids (1 is the longest, 8 the shortest) and colors indicate shared k-mers.

The same four simulated samples as in the F threshold test (Fig. 4) were used to assess the effect of different C thresholds: 95%, 80% and 65% (Table S2). F threshold was set to 80%. As expected, false positives (clusters that did not contain any plasmids present in the sample) emerged at higher clustering thresholds due to similar plasmids considered as two or more separate hits. At lower thresholds, these plasmids formed a single cluster and were thus considered a correct hit. Very low clustering thresholds can result in aberrant co-clustering of separate plasmids present in the sample, which results in losing some of the true positives. Therefore, we decided to set PlasmidSeeker default clustering threshold C to 80%.

Analysis of Escherichia coli WGS samples with PlasmidSeeker compared to plasmidSPAdes

To further evaluate the performance of PlasmidSeeker on real WGS data, we randomly chose three E. coli WGS samples (named EC1, EC2 and EC3) with sequence type ST131 from a previously published dataset (Roosaare et al., 2017) (see Materials and Methods) and used PlasmidSeeker and plasmidSPAdes to identify plasmids (Table 2; Table S3). No plasmids have been previously annotated for these samples, thus true positives were unknown. Both tools assume that plasmid copy numbers differ significantly from the isolate. PlasmidSPAdes assembles reads, finds the read coverage for each contig and separates putative plasmid contigs by their lower or higher read coverage compared to chromosomal contigs. To validate plasmidSPAdes results, we used the same approach as its authors, running a blastn search against the NCBI database of nonredundant nucleotides using the longest contig of each putative plasmid. All plasmid matches found by blastn that had over 95% identity to the longest contig and had also been detected by PlasmidSeeker were considered as detected by both tools. PlasmidSeeker could not be validated by blastn as it only shows the detected reference plasmids in the output. Only the top hit of each cluster is shown for PlasmidSeeker.

Table 2 Performance of PlasmidSeeker and plasmidSPAdes on E. coli WGS samples.

	PlasmidSeeker	PlasmidSPAdes	
Sample	Reference plasmid	% k-mers found	Plasmid length	Copy number	Contigs (max contig)	Plasmid length	Blastn hitcoverage/identity	Copy number	
EC1	pEC867_3	94.12	4,074	67.64	1 (4,129)	4,129	plasmid 100/99	66.45	
pEC743_1	80.38	111,851	2.82					
				164 (20,911)	197,083	plasmid 89/99	2.56	
				1 (29,236)	29,236	genomic (97/99)	1.64	
				2 (28,336)	28,336	genomic (100/100)	1.65	
EC2	pJJ2434_1	92.39	126,302	2.61	30 (30,273)	181,896	plasmid (100/99)	2.99	
p2PCN033	93.95	4,086	0.33	1 (4,137)	4,137	plasmid (99/100)	0.32	
pVR50I	87.75	1,552	0.22	1 (1,607)	1,607	plasmid (100/99)	0.18	
C227-11 plasmid	87.16	2,954	2.67					
pJJ1886_2	99.61	5,167	79.56					
pJJ1886_3	100.00	5,631	41.78					
				1 (2,156)	2,156	plasmid (100/100)	0.1	
				7 (5,277)	11,554	genomic (100/99)	8.3	
EC3	pEC732_5	100.00	1,549	482.29					
pVR50E	95.74	5,164	89.14					
pHUSEC41-3	90.70	7,930	52.86					
pNJST258N4	97.79	14,249	7.07					
pSH146_87	88.68	86,586	1.86					
				45 (80,840)	229,182	plasmid (74/99)	3.02	
				24 (9,730)	39,730	genomic (100/100)	2.56	
				3 (10,125)	10,833	genomic (100/99)	5.3	
Note:

Plasmids found by both tools are on the same row.

In case of sample EC1, both PlasmidSeeker and plasmidSPAdes detected the same plasmids with similar copy number (Table 2). PlasmidSPAdes detected another large plasmid. Blastn showed that the best match for this plasmid had only 89% coverage, indicating that the closest reference plasmid was not very similar, which may have been the reason why PlasmidSeeker did not detect it. Two of the additional components found by plasmidSPAdes originated from the E. coli genome according to blastn.

In EC2, three plasmids with similar copy numbers were detected by both programs. Three additional plasmids were identified by PlasmidSeeker and one by plasmidSPAdes.

Interestingly, there was no overlap between the results in case of EC3. PlasmidSPAdes identified a single large plasmid, two other components originating from the genome. PlasmidSeeker identified five plasmids. Remarkably, one of the putative small plasmids (pEC732_5, 1,549 bp) had an estimated copy number as high as 482. To validate this find, we ran a blastn search against the WGS reads of EC3 using the reference sequence of pEC732_5. There were 91,354 matching 101 bp reads with 100% identity and 100% coverage, indicating a putative copy number of 397 (median genome coverage of the isolate was estimated by PlasmidSeeker to be 15). Another blastn search was performed with pEC732_5 against all the contigs assembled by plasmidSPAdes. No significant similarity was found, which indicated that plasmidSPAdes had probably not assembled reads matching pEC732_5 reference sequence. We analyzed another putative high-copy plasmid, pVR50E, in the same way. It had 51,223 matching WGS reads and 21 matching small contigs (100–242 bp) assembled by plasmidSPAdes. Median copy number of these contigs was 88.16, which is very similar to the copy number estimated by PlasmidSeeker (89.14 copies). These contigs were not designated as part of a putative plasmid by plasmidSPAdes.

Overall, PlasmidSeeker and plasmidSPAdes seem to complement each other. PlasmidSeeker was unable to detect putative plasmids which either had very low copy numbers or were not very similar to reference plasmids. PlasmidSPAdes failed to detect some of the putative plasmids with high-copy numbers.

Discussion

In this work, we introduced a k-mer based plasmid detection tool called PlasmidSeeker. We showed that the PlasmidSeeker can detect the presence of plasmids with known sequences both from simulated and real bacterial WGS data without read assembly. However, our approach is currently suitable only for WGS samples. Considering shotgun metagenomics sequencing data, we do not know which organisms are present in the sample and thus cannot know if the detected k-mers originate from plasmids or from unknown organisms. Compared to next-generation sequencing technologies, optical DNA mapping might be better suited to detect plasmids from metagenomics samples, as it visualizes the intact plasmid sequence instead of breaking it up into short reads as sequencing inevitably does (Nyberg et al., 2016).

Often, PlasmidSeeker detected multiple plasmids with identical copy numbers even as only one of the plasmids was present in the sample. This is due to many plasmids in our database being very similar to one another. To address this, we clustered the detected reference plasmids based on the overlap coefficient. We considered using the Jaccard similarity index instead, but we did not find it feasible because of cases in which a small plasmid shared most of its k-mers with a large plasmid but were not similar according to the Jaccard index. Additionally, we did not create clusters in the database building step, as the clusters are dependent on the plasmids predicted to be in the sample. For example, this would be a problem for the hypothetical scenario in which two small plasmids are similar to parts of a large plasmid, but not to each other. If the large plasmid is present in the sample, k-mers of the small plasmids will be found as well, and all three should be clustered together. If only the two small plasmids are detected in the sample, we should not cluster them as they are not similar. Therefore, clustering is done after the search process.

As PlasmidSeeker is based on detecting plasmid k-mers from the sample, choosing an optimal k-mer length when building the database is of high importance. Short k-mers (k < 16) are abundant in most bacteria and plasmids, limiting their specificity. Long k-mers (k > 20), while specific, may remain undetected due to sequencing errors and mutations in plasmids (differences compared to reference plasmids), causing a loss in sensitivity (Fig. 2). Our tests indicated that the optimal k-mer length is 20 bp.

PlasmidSeeker assumes that the bacterial chromosome and plasmid sequences have different copy numbers. It should be noted that the culturing process and library preparation can influence the copy number of bacterial plasmids (Turner, 2004) and thereby the performance of PlasmidSeeker.

PlasmidSeeker can also be used to identify purified plasmids. For this, the sequence of a reference bacterium is not needed, and plasmid coverages are given in the output instead of copy numbers.

PlasmidSeeker has the following requirements and limitations: Plasmids can be identified only from bacterial WGS reads (a sample containing only an isolated bacterium and its plasmids). Shotgun metagenomics samples are not yet supported.

At least 5× coverage for both the isolate and plasmids is required to detect a plasmid, otherwise many of their k-mers might not be detected (Fig. 3). With higher sequencing depth of the sample, even plasmids with copy numbers lower than one can be detected (Table 2).

PlasmidSeeker can only identify plasmids similar to reference sequences in the database (Fig. 2).

Conclusion

We have developed a novel tool to detect plasmids from bacterial WGS data without read assembly. Using simulated and real bacterial WGS samples, we showed that PlasmidSeeker can detect known plasmids and accurately estimate their copy numbers. PlasmidSeeker is suitable to use as a first step in the analysis of plasmid content and it complements tools that assemble reads and are thus able to detect novel plasmids.

Supplemental Information

Supplemental Information 1 K-mer threshold effects on results in real data (k = 20).

Click here for additional data file.

Supplemental Information 2 Clustering threshold.

Click here for additional data file.

Supplemental Information 3 PlasmidSeeker and plasmidSPAdes results on real E. coli samples.

Click here for additional data file.

Additional Information and Declarations

Competing Interests

Author Contributions

Data Availability

The authors declare that they have no competing interests.

Märt Roosaare conceived and designed the experiments, performed the experiments, analyzed the data, prepared figures and/or tables, authored or reviewed drafts of the paper, approved the final draft.

Mikk Puustusmaa conceived and designed the experiments, performed the experiments, analyzed the data, prepared figures and/or tables, authored or reviewed drafts of the paper, approved the final draft.

Märt Möls analyzed the data, contributed reagents/materials/analysis tools, approved the final draft.

Mihkel Vaher conceived and designed the experiments, authored or reviewed drafts of the paper, approved the final draft.

Maido Remm authored or reviewed drafts of the paper, approved the final draft.

The following information was supplied regarding data availability:

The PlasmidSeeker database is available http://bioinfo.ut.ee/plasmidseeker and at FigShare: Roosaare, Märt (2017): PlasmidSeeker database k = 20. figshare. https://doi.org/10.6084/m9.figshare.5497972.v1.

PlasmidSeeker source code is available at GitHub: https://github.com/bioinfo-ut/PlasmidSeeker.

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
