# Peer review of "PlasmidSeeker: identification of known plasmids from bacterial whole genome sequencing reads"

_PeerJ, doi:10.7717/peerj.4588_

## Round 0.1 · original submission · Minor Revisions

· Academic Editor

Minor Revisions

Dear Dr. Roosaare and colleagues:

I have now received two reviews of your work, and I am happy to relay to you that both are fairly positive. Please take a look at the concerns raised by the reviewers and address them accordingly. I believe that your work will be accepted for publication after a minor revision. I am curious of your response about selected read length as it pertains to practical data, such as MiSeq. Are your results robust to a variety of read-lengths? I look forward to your revision…good luck!

-joe

Reviewer 1 ·

Basic reporting

The authors present a novel tool, PlasmidSeeker, which examines the co-occurence of k-mers between a query sequence, bacterial host and a plasmid database. This is done directly from raw reads, thus giving higher sensitivity but also saves the user for the major computational task of doing assembly.

Experimental design

As described by the authors, a method based solely on k-mers requires a database that is highly similar to the reference sequences. Here the sensitivity and specificity is tested with artificial sequence, mutated with SNP’s in different amounts. Plasmids are however also known to be extremely plastic, meaning that they quite quickly exchange their genomic information by switching out entire gene islands, as described by M. Couturier et al. (Couturier et al. 1988). Which is part of the reason why PlasmidFinder is based on replicons, as these are the constant parts of the plasmid (Carattoli et al. 2014). Another benchmark is thus needed, where entire islands of genes are changed.

Besides the mapping of k-mers to a reference database, the authors also examine the copy-number difference between chromosomal k-mers and plasmid k-mers, where these are assumed to be different. The copy number of plasmids may be influenced by the library preparation used, and might therefore have impact on the detection and be worth mentioning (Turner n.d.).
Overall, PlasmidSeeker seems to have great potential, and might help in outbreak investigations to gain a deeper insight on plasmid spread.

Carattoli, Alessandra et al. 2014. “In Silico Detection and Typing of Plasmids Using PlasmidFinder and Plasmid Multilocus Sequence Typing.” Antimicrobial Agents and Chemotherapy 58(7):3895–3903. Retrieved February 8, 2018 (http://www.ncbi.nlm.nih.gov/pubmed/24777092).
Couturier, M., F. Bex, P. L. Bergquist, and W. K. Maas. 1988. “Identification and Classification of Bacterial Plasmids.” Microbiological Reviews 52(3):375–95. Retrieved February 8, 2018 (http://www.ncbi.nlm.nih.gov/pubmed/3054468).
Turner, Paul E. n.d. “Phenotypic Plasticity in Bacterial Plasmids.” Retrieved February 8, 2018 (https://www.ncbi.nlm.nih.gov/pmc/articles/PMC1470877/pdf/15166133.pdf).

Validity of the findings

No comment

·

Basic reporting

The text is clearly written and provides sufficient background information and references.

A Makefile (or similar process file) could be referenced such that the reader could easily recreate the analysis in the paper,

Experimental design

The design seems reasonable, and the necessary parameter validation steps appear to have been performed.

The purpose of the tool is well defined and targets an important research question i.e. robustly identifying plasmid sequences in WGS data.

I would ask why 80bp was used for read simulation? A more realistic length would match that of MiSeq data i.e. 250-300bp. Does read length have any effect on k-mer frequencies and therefore the accuracy of results?

Validity of the findings

The results are well presented and the discussion and conclusions seem fair.

Additional comments

Line 86: claims that a reference genome from the same genus would be sufficient. This might be quite genus specific, there can be quite a high level of genome diversity in some genera. I don't think any evidence has been presented to support this?

Line 133: can the "in house script" be made available? Maybe through github or in supplementary. It would help the reader reproduce the analyses


A general comment, always write the species name in full the first time, and then subsequently abbreviate. E.g. Escherichia coli written in full at line 114, 189 and 211. Also check other species names.

---

## Round 0.2 · accepted · Accept

· Academic Editor

Accept

Dear Dr. Roosaare and colleagues:

Thank you for revising your manuscript and resubmitting in a timely fashion. I believe your revision has met the concerns raised by both reviewers, therefore I am accepting your work for publication in PeerJ. Congratulations!

I anticipate that PlasmidSeeker will be a valuable resource for the bioinformatics and genomics community. Thanks for publishing with PeerJ, and I look forward to seeing your work in print!

Best,

-joe